# Contribution of Glacier Runoff during Heat Waves in the Nooksack River Basin USA

**Mauri S. Pelto** [1],*, **Mariama Dryak** [2], **Jill Pelto** [3], **Tom Matthews** [4] **and L. Baker Perry** [5]

1 Environmental Science, Nichols College, Dudley, MA 01571, USA
2 Environmental Science, Shenandoah University, Winchester, VA 22601, USA; mdryak@su.edu
3 North Cascade Glacier Climate Project, Dudley, MA 01571, USA; pelto.jill@gmail.com
4 Geography, Kings College London, London WC2R 2LS, UK; tom.matthews@kcl.ac.uk
5 Geography and Planning, Appalachian State University, Boone, NC 28608, USA; perrylb@appstate.edu
* Correspondence: mspelto@nichols.edu

**Abstract:** The thirty-eight-year record (1984–2021) of glacier mass balance measurement indicates a significant glacier response to climate change in the North Cascades, Washington that has led to declining glacier runoff in the Nooksack Basin. Glacier runoff in the Nooksack Basin is a major source of streamflow during the summer low-flow season and mitigates both low flow and warm water temperatures; this is particularly true during summer heat waves. Synchronous observations of glacier ablation and stream discharge immediately below Sholes Glacier from 2013–2017, independently identify daily discharge during the ablation season. The identified ablation rate is applied to glaciers across the North Fork Nooksack watershed, providing daily glacier runoff discharge to the North Fork Nooksack River. This is compared to observed daily discharge and temperature data of the North Fork Nooksack River and the unglaciated South Fork Nooksack River from the USGS. The ameliorating role of glacier runoff on discharge and water temperature is examined during 24 late summer heat wave events from 2010–2021. The primary response to these events is increased discharge in the heavily glaciated North Fork, and increased stream temperature in the unglaciated South Fork. During the 24 heat events, the discharge increased an average of +24% (±17%) in the North Fork and decreased an average of 20% (±8%) in the South Fork. For water temperature the mean increase was 0.7 °C (±0.4 °C) in the North Fork and 2.1 °C (±1.2 °C) in the South Fork. For the North Fork glacier runoff production was equivalent to 34% of the total discharge during the 24 events. Ongoing climate change will likely cause further decreases in summer baseflow and summer baseflow, along with an increase in water temperature potentially exceeding tolerance levels of several Pacific salmonid species that would further stress this population.

**Keywords:** glacier ablation; North Cascade Range; climate change; salmon; glacier mass balance; heat wave

## 1. Introduction

Climate observations in the Pacific Northwest (United States) show an accelerated warming for the 1970–2012 time periods of approximately 0.2 °C per decade [1]. Analysis of key components of the alpine North Cascade hydrologic system indicate significant changes in glacier mass balance, terminus behavior, alpine snowpack, and alpine streamflow from 1950–2015 [2,3]. Glacier runoff is of particular importance to aquatic life late in the summer when other water sources are at a minimum, raising minimum streamflow and reducing maximum temperatures [4]. Contributions from groundwater, precipitation, and non-glacier snowmelt reach a minimum after 1 July [4]. Whereas annual glacier runoff peaks during the late summer and is highest in warm, dry summers and lowest during wet, cool summers [5].

Watersheds in the Pacific Northwest are comprised of pluvial streams that experience peak flow in winter due to the winter storm events [6], nival streams that peak in the May and June due to high snowmelt, and glacially fed streams which peak in July and August

during peak glacier melt [5–8]. A comparison of hydrographs from glaciated and unglaciated basins indicates a similar progression through June when runoff is dominated by non-glacier snowmelt, followed by increasing divergence as glacier runoff minimizes declines in glaciated watersheds until October when the hydrographs converge again [7,8]. The loss of a glacier from a watershed reduces streamflow primarily during late summer minimum flow periods [5,8,9]. The volume of glacier runoff is the product of surface area and ablation rate [2]. Glacier volume loss contributes to changes in streamflow, leading to an increase in overall streamflow if the rate of volume loss is sufficiently large [7], or a decline in streamflow if the area of glacier cover declines sufficiently to offset any increase in ablation rate [2]. It is evident that glaciers have a substantially larger role than the area they cover in August based on the identifiable glacier fingerprint on hydrographs for a watershed where glacier cover exceeds 1% of total watershed area [8]. The amount of summer runoff generated per unit area in the Nooksack River was 0.036 $m^3s^{-1}km^{-2}$, in the unglaciated South Fork Nooksack (SFN) discharge was 0.045 $m^3s^{-1}km^{-2}$, increasing to 0.312 $m^3s^{-1}km^{-2}$ in the heavily glaciated North Fork Nooksack (NFN). This represents nearly a seven-fold increase in runoff from glacier versus non-glacier areas in the Nooksack Basin [2].

Climate change is altering late summer streamflow in the North Cascades. There has been a coherent shift toward earlier runoff in snow fed basins across the western US, including a 10–30-day earlier date of the center of mass for annual flow for each water year [10]. A reduction in summer streamflow in six North Cascade basins from 1956–2006 has been observed [2]. In the North Cascades glacier volume loss has contributed up to 6% of the total August–September streamflow [11]. The loss in glacier area in the North Cascades and British Columbia is greater than the increased rate of ablation, as a result peak runoff in these same regions has been reached and the dominant ongoing change in glacier runoff is a decline in summer streamflow due to glacier area reductions [2,7,12].

Thermal regimes in streams reflect the balance of numerous physical processes that cause heating or cooling. Rates of warming in the Pacific Northwest's rivers have been highest during the summer, an increase of 0.17 °C per decade [4]. Air temperature was the dominant factor in both long term and inter-annual variability for Pacific Northwest rivers [4]. Discharge and air temperature appear additive and the seasonal variation in stream warming rates is determined by the degree of concert between these two variables [4]. For example, the largest warming trend during the summer resulted from the effects of the largest air temperature increases added to the largest river discharge decreases. This is further supported by Luce et al. [13] who identified a pattern where water temperature in cold streams had low sensitivity to air temperature, while warm streams had a tendency for higher sensitivity to air temperature.

An important impact of changing glacier runoff in the Nooksack River is the stress of warming stream temperatures on salmon populations [13,14]. Temperature thresholds for changes in fish communities in the Fraser River region of British Columbia were noted as 12 °C and 19 °C [14]. The reduction of the glacial melt component augmenting summer low flows is already resulting in more low-flow days in the North Cascade region as has occurred other alpine regions with small glaciers [12,15]. In the Skykomish River from 1950–2013, there were 230 days during the summer melt season with discharge below 10% of mean annual flow (14 $m^3s^{-1}$); of these, 99% (228 days) had occurred since 1985 [12]. Of great concern for aquatic life is the occurrence of extended periods of low flow [14] that have increased in frequency.

Climate change is a growing threat that has caused and will cause increases in winter flow, earlier spring snowmelt, decreased summer baseflow, and increased maximum summer water temperature in North Cascade watersheds [14]. Without mitigating steps, climate change will increase the frequency of low flow conditions and water temperatures that exceed the salmon tolerance levels. The impact is most acute during summer heat waves that result in minimum flow conditions coincident with maximum stream temperatures. This research identifies the specific response in glaciers ablation, glacier runoff and the resultant evolving water temperature threat during summer heat waves in the

Nooksack River Basin. In this study, heat events are identified as any period of five or more consecutive days where the average daily temperature observed at the Middle Fork Nooksack SNOTEL site (MFN) exceeds 14 °C, and precipitation is less than 3 mm for the entire period. A shorter heat event due to lagging responses would not yield a robust measure of streamflow response. The temperature threshold is simply chosen based on relation to locally identified heat waves. More than 3 mm of precipitation could influence discharge and complicate understanding the contrasting response of SFN and NFN. This is accomplished by monitoring ablation and runoff directly from Sholes Glacier and examining the simultaneous United States Geological Survey (USGS) discharge and stream temperature record in the two of the three principal forks of the Nooksack River, having varying amounts of glacier cover, for 24 summer heat waves. This includes the most intense heat wave the region has experienced occurring at the end of June 2021. The exceptional nature of the June 2021 heat wave is identified using summer air temperature reanalysis using ERA5.

## 2. Study Area

The Nooksack River consists of the North, South, and Middle Fork which combine near Deming to create the main stem Nooksack River. The Nooksack River empties into Birch Bay near Bellingham, Washington. The Nooksack River Basin is a hybrid basin with the various sub-basins dominated by pluvial, nival, and glacial runoff contributions, resulting in differing seasonal timing of maximum discharge, reducing the magnitude and duration of the summer minimum flow period. The USGS has gaging stations on each of the three main forks and the main stem of the Nooksack Basin. There are no significant reservoirs or flow diversions upstream of the gaging locations. From October-March is a storage period characterized by precipitation exceeding discharge, whereas April-August is a period of excess runoff release [6,16]. In the Nooksack River basin, glacier runoff supplied 10–20% of summer streamflow in the late 20th century [16]. The primary focus is on glacier runoff for the North Fork Nooksack River with 6.1% glacier cover above the gaging station (Figure 1). The Nooksack River system is home to five species of Pacific salmon including: Chinook, Coho, Pink, Chum, and Sockeye, with Chinook listed as Threatened under the Endangered Species Act (ESA) [14]. In the last two centuries the numbers of salmon that return to spawn in the Nooksack watershed have greatly diminished due to substantial loss of habitat primarily from human-caused alteration [14].

Thirty-seven years of mass balance work in the basin identify glacier ablation that yields an average of 11 $m^3s^{-1}$–12 $m^3s^{-1}$ from July–September [3]. This is 10–20% of the total summer discharge of the Nooksack River at Ferndale, Washington, depending on the specific year. The glaciated area coverage in 2015 was 6.1% in North Fork Nooksack River (NFN) basin, 0% in South Fork Nooksack River (SFN) basin, and 1.1% in the Nooksack River basin at Ferndale (Figure 1). This difference in glacier covered area allows assessment of the impact of glaciers on stream discharge and temperature.

The NFN is a 65-km long tributary, with salmon habitat extending to the base of Nooksack Falls (Figure 1). From 1985 to 2017 mean July-September discharge is 25.9 $m^3s^{-1}$. In 2015, the NFN watershed had a glacier area of 16.9 $km^2$, with 12.3 $km^2$ of that glacier area located above the USGS gage site. On the NFN at Nooksack Falls there is a run of river hydropower plant constructed in 1906 that is rated at a production of 3.5 MW. There is no reservoir for this plant, just a low weir diverting water into a penstock above the falls. The 1.25 km long penstock returns diverted water to the river below the falls.

The SFN is a 57-km long tributary with salmon habitat extending 52 km upstream of the junction of with the Nooksack River. From 1985 to 2017 mean July–September discharge is 8.8 $m^3s^{-1}$. There are currently no glaciers in the watershed. There are no hydropower plants in the basin.

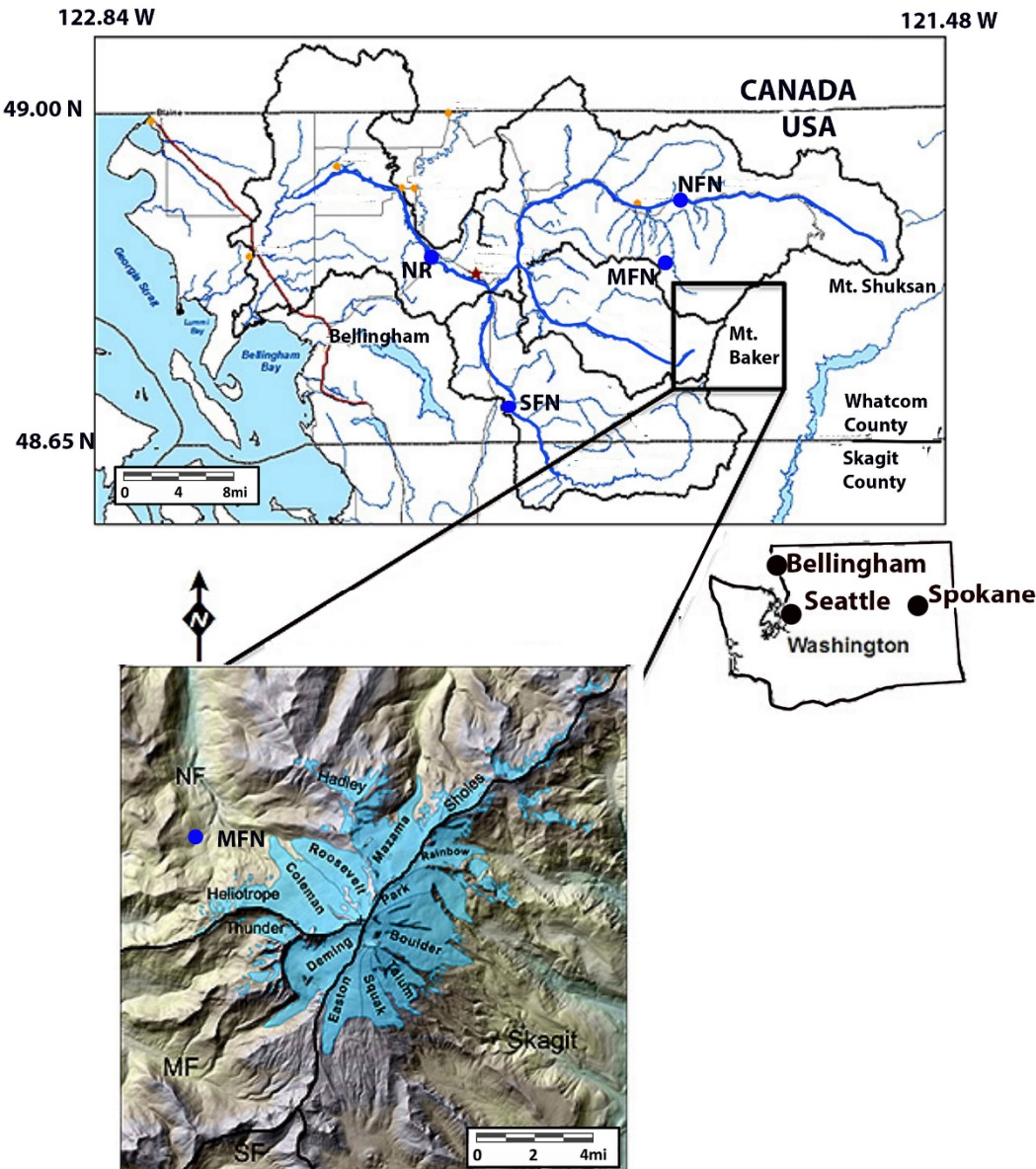

**Figure 1.** Map of Mount Baker glaciers and the Nooksack River Watershed. USGS gage locations for the Nooksack River (NR), North Fork Nooksack River (NFN), and South Fork Nooksack River (SFN) are indicated with blue dot. The Middle Fork Nooksack SNOTEL station (MFN) also indicated with blue dot, on inset also. Yellow dots mark cities, Red line is the I−5 highway and the red star is the Nooksack Indian Tribal Center.

From 1950–1980 the areal extent of glaciers in the NFN basin increased, with all Mount Baker glaciers advancing [17,18]. Since 1980, all glaciers in the basin have retreated significantly, with the retreat accelerating since 2000 [17,18]. On Mount Baker the average glacier retreat was 430 m during 1979–2015 [19]. Mass balance measurements indicate the cumulative loss as −17.3 m water equivalent (w.e.), equivalent to 20–30% of glacier volume lost during 1984–2015 [3].

## 3. Methods and Data Sources

### 3.1. Glacier Mass Balance

The North Cascade Glacier Climate Project (NCGCP) has monitored the annual mass balance during 1984–2021 on Lower Curtis and Rainbow Glacier in the Baker Lake watershed adjacent to the NFN and from 1990–2021 mass balance on Sholes Glacier in the NFN

and Easton Glacier in the Baker Lake watershed [3,18,20]. Rainbow Glacier, which abuts Sholes Glacier, and Easton Glacier 8 km south in the Baker Lake Watershed are two of the 42 reference glaciers for the World Glacier Monitoring Service (WGMS). Accumulation and ablation measurements are completed yearly during the summer on each glacier at a density of over 100 pointskm$^{-2}$, and changes in glacier area are assessed every three to five years. The program relies on consistent methods applied to the same network of data points on the glacier's each year [20], with an uncertainty of 0.15 ma$^{-1}$ falling in the typical range [21,22]. Direct measurement of ablation is accomplished using ablation stakes, changes in snow depth from repeat probing measurements and from snowline migration [19]. Ablation stakes are distributed across the entire elevation span of the glacier [3]. Stake measurement error over the shorter periods of observation determined here and in the Swiss Alps is 0.05 m [22,23], but less than 0.11 m to 0.14 m errors reported for annual stake observations [21,24]. These data, including the specific glacier area, are reported annually to the WGMS. Sholes Glacier had a mean elevation of 1840 m in 2015, while the mean elevation of glaciers in the NFN above the USGS gaging station was 1820 m in 2015 [19].

Overall Sholes Glacier has had a mass balance of −24.9 m w.e. during 1990–2021, this is a substantial loss for a glacier that averages 40–60 m in thickness [19]. The highest rate of loss occurred from 2013 to 2021 with a 13 w.e. loss. The correlation in annual mass balance is from 0.96–0.98 between Sholes Glacier and Lower Curtis, Rainbow and Easton for 1990–2017, indicating the nearly identical response to annual climate conditions [3,20]. Glacier ablation measurements occurring simultaneously with discharge measurements below the glacier provide independent measures of glacier runoff. Here we report on observations during specific time periods that overlap with heat wave periods at ablation stakes.

The degree day function (DDF) is the most common means of modelling ablation on glaciers [25]. In this study both daily ablation and multi-day ablation observations are used to identify how much glacier runoff is produced. All daily ablation measurements from Easton, Sholes, and Rainbow Glacier completed during the summer have been used in combination with daily mean temperatures at MFN to derive a degree day factor (DDFs and DDFi) for daily snow and ice ablation respectively. In this study a specific heat event DDF is derived for snow and ice ablation for days where the temperatures average is 13 °C or above at MFN. This is 1 °C below the heat event threshold and expands the data set while maintaining the high temperature selection. The DDF Equation (1) is based on the average daily ablation (DA) at multiple sites between 1700 and 1900 m and the daily mean temperature (DT) at MFN. Neither ablation nor temperature is adjusted to the specific stake elevation. If there was a greater range in elevation of the ablation sites a lapse rate would be appropriate. Daily ablation measurements have been completed on 48 separate days, yielding 178 location specific observations when temperatures have exceeded this threshold. Of the 24 heat events we have collected ablation data throughout 10 of them. These data are used to generate a DDF model for ablation conditions during warm, dry periods.

$$\text{DA} = (\text{DDFs} \times \text{DT}) \text{ or } (\text{DDFi} \times \text{DT}) \tag{1}$$

### 3.2. USGS Stream Data

Both daily and monthly records of stream temperature and discharge are available from USGS stations at: North Fork Nooksack River at Glacier, South Fork Nooksack River at Saxon Bridge and Nooksack River at Ferndale and Cedarville [16,19]. Table 1 indicates the gage location characteristics, data type, and period of record utilized. This allows comparison of stream response to specific weather conditions and comparison between basins. The stream temperature records from the USGS only exist since 2008; hence no temporal trend analysis is performed.

**Table 1.** USGS stations characteristics and data records utilized.

| Basin | USGS Station ID | Mean Elevation m a.s.l. | Basin Area km$^2$ | Glacier Cover % | Discharge Records | Stream Temperature Records |
|---|---|---|---|---|---|---|
| Nooksack | 12213100 | 800 | 2036 | 1.1 | 1970–2013 | None |
| SF Nooksack | 12210000 | 914 | 334 | 0 | 2008–2013 | 2008–2013 |
| NF Nooksack | 12205000 | 1311 | 272 | 6.1 | 1950–2013 | 2008–2013 |

*3.3. SNOTEL Data*

The United States Department of Agriculture-SNOTEL program has two stations in the Nooksack Basin that provide daily air temperature and precipitation. The Middle Fork Nooksack (MFN) station provides a consistent measure of hourly temperature and precipitation at an elevation 300 m below the glacier and 9 km west of Sholes Glacier, while Wells Creek is 600 m below the glacier and 6 km northwest.

*3.4. ERA5 Data*

There is no weather station above 1600 m in the area, below all but the lowest areas of a few glaciers. To understand the temperatures at higher elevations required use of an ERA5 dataset. Three-hourly air temperatures and geopotential height on pressure levels were obtained for the ablation season May–September from May 1979 to July 2021 from the 0.25 × 0.25° ERA5 dataset, the latest and highest resolution reanalysis produced by the European Centre for Medium Range Weather Forecasting [26]. Both fields were interpolated to −121.8° E, 48.8° N. The geopotential height, indicating the elevation of the pressure levels, was then used as the vertical coordinate to linearly interpolate temperature to the 3000 m contour. Daily maximum temperatures were computed from these interpolated data. The temperature lapse rates were defined as the slope coefficient in a regression of air temperature against the corresponding geopotential height [27]. We summarized the daily mean lapse rate as the arithmetic average of these (eight) slope coefficients computed on the three-hourly data. We caution that our use of ERA5 reanalysis in place of direct observations is a source of uncertainty, however given the lack of a long or consistent temperature record at elevation, this provides the most comparable data record for evaluating the significance of specific heat events. A good general agreement between station-based estimates of temperature and ERA5 in Western U.S. region, overlapping our region of study has been noted [28]. The highest seasonal correlation between stations and ERA5 for environmental lapse rate was 0.7 during the summer [28]. The highest correlations for specific temperature differences between stations and ERA was 0.9 for maximum temperatures [28]. This indicates that maximum temperatures in summer are one of the most reliable products of ERA5 in the Western US [28]. We also note that ERA5 is used here only to identify periods of highest temperatures, hence any mean biases will not affect our conclusions.

## 4. Results

*4.1. Glacier Ablaiton*

A few specific examples are reviewed below. In 2014 measurements of ablation daily during the 27 July–7 August period at a series of 12 stakes on the Sholes Glacier indicated a mean ablation rate for snow of 0.055 m w.e.d$^{-1}$. The ice melt was of the same thickness as noted for snow, but because of the greater density the water equivalent loss is higher, it indicated a mean ablation of 0.75 m w.e.d$^{-1}$. Between 29 July and 4 August 2015 measurements of ablation at a series of 6 stakes on the Sholes Glacier indicated a mean ablation rate for snow of 0.057 m w.e.d$^{-1}$ and for ice of 0.078 m w.e.d$^{-1}$. In 2016, ablation measurement during the 12–21 August period at a series of 12 stakes on the Sholes Glacier yielded average snow ablation of 0.048 m w.e.d$^{-1}$ and ice ablation of 0.070 m w.e.d$^{-1}$. In 2020, from 29 July to 5 August mean snow ablation at six stakes was 0.055 md$^{-1}$ w.e.

*4.2. Ablation Modelling*

From 1990 to 2018, daily ablation measurements on Sholes Glacier and daily mean temperature reported at the Middle Fork Nooksack SNOTEL station, are utilized to generate a DDF for ablation. This model is generated from 148 days of observations of both ablation and air temperature (Figure 2). The focus on the observations for this study is specific daily to weekly observations of ablation during heat waves at the network of stakes spread across the glacier (Figure 3).

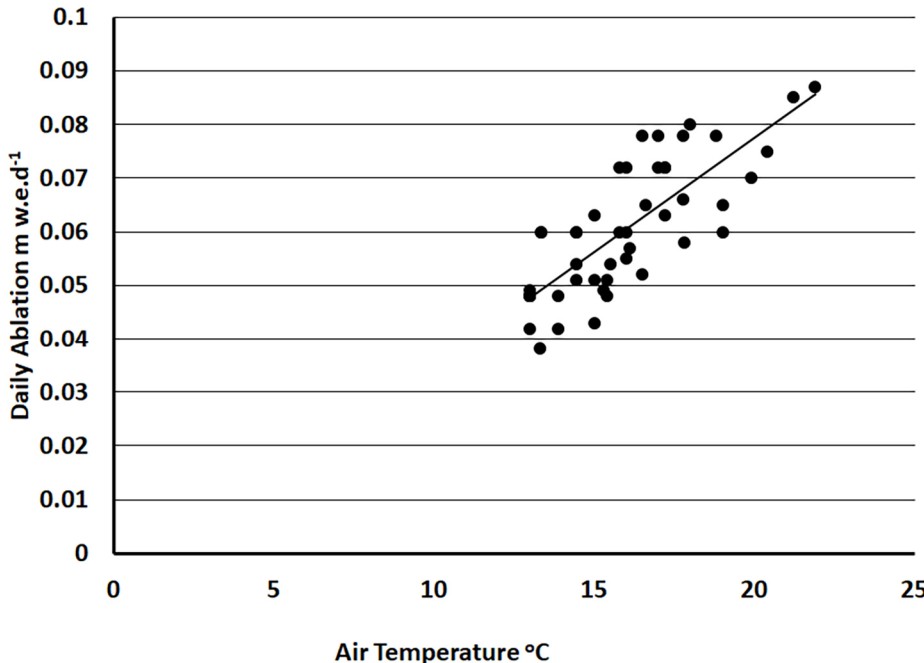

**Figure 2.** Daily mean air temperature at the Middle Fork Nooksack SNOTEL site and daily snow ablation measured on Sholes Glacier. The DDFs are derived from the linear regression slope coefficients.

The correlation coefficient between observed ablation and daily mean temperature is 0.82, for the entire ablation data set regardless of mean daily temperature. The overall DDFs for snow is 0.0035 m w.e. $°C^{-1}d^{-1}$. For ice, the DDFi is 0.0053 m w.e. $°C^{-1}d^{-1}$ [20].

This is similar to the reported relationship for nearby South Cascade Glacier during the 2003–2007 period; for snow was 0.0038 m w.e. $°C^{-1}d^{-1}$ and for ice was 0.0054 m w.e. $°C^{-1}d^{-1}$ [29]. Both ablation rates and DDF relationship in the limited elevation range of North Cascades glaciers have been found to be consistent from glacier to glacier [3,12,20,29]. The DDFs for Rainbow Glacier and Easton Glacier each with extensive mass balance records are between 0.0033 and 0.0039 m w.e. $°C^{-1}d^{-1}$. Both the annual balance and DDF relationships indicate it is reasonable to utilize Sholes Glacier ablation as a proxy for ablation on other glaciers in the watershed [3,19]. Sholes Glacier's mean elevation is also within ~20 m of the mean elevation of glaciers in the watershed in 2015.

For heat waves we have derived a separate DDF relationship based only on the 48 days when the average temperature at the Middle Fork Nooksack SNOTEL station exceeded 13 °C and we were measuring ablation. During heat waves, the DDF relationship changes yielding higher values with DDFs snow of 0.0043 m w.e. $°C^{-1}d^{-1}$. For ice, the DDFi is 0.0067 m w.e. $°C^{-1}d^{-1}$.This underscores the observation on other glaciers that incorporating weather types into a degree day model improves performance [30]

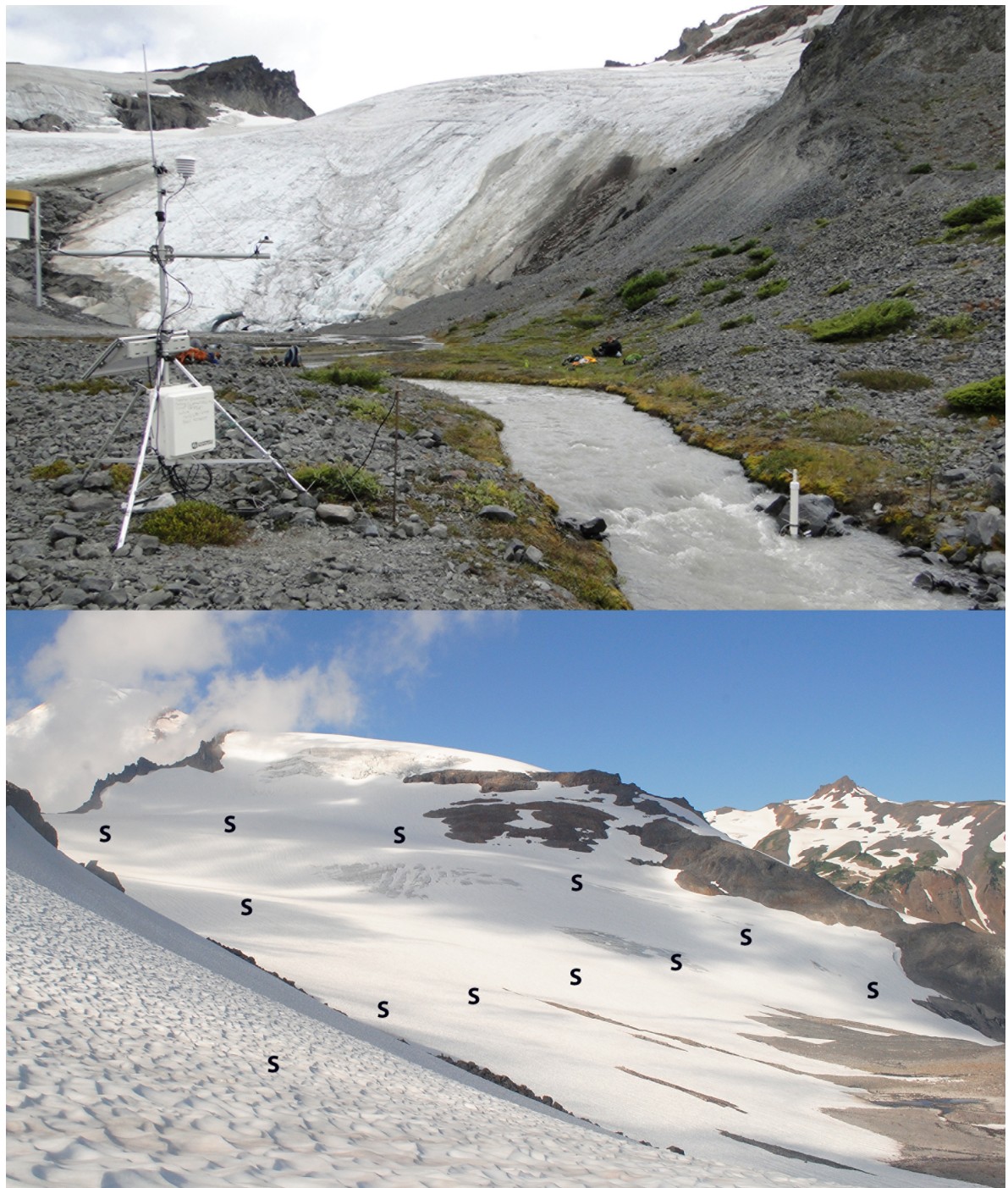

**Figure 3.** Limited retained snow cover on 8 August 2015 on Sholes Glacier above, with the discharge and weather station in foreground. Below is the stake network on Sholes Glacier annotated on this 2014 image.

The daily air temperature is scaled by the DDF to provide a daily value for ablation that can in turn be multiplied by the area of glacier ice and glacier snow to calculate the volume of runoff from Sholes Glacier and from all NFN glaciers. The model is further validated by comparison with periods of detailed ablation field observations (Table 2), yielding a mean daily ablation rate within 10% of observed ablation rates. Overall glacier runoff is the sum of the product of DDFs and snow-covered area, and DDFi and ice-covered area (Table 3). For each of the periods in Table 2, field work was completed during or within

three days of the heat event allowing mapping of the snow and ice area. The area of snow and ice is a significant variable from year to year and through the melt season (Figure 3).

**Table 2.** Ablation rates determined from field measurements and degree day modeling on Sholes Glacier during portions of the 2012–2020 melt seasons.

| Dates | Snow Ablation Rate-Measure (m w.e.d$^{-1}$) | Snow Ablation Rate-Model (m w.e.d$^{-1}$) | Ice Ablation Rate-Model (m w.e.d$^{-1}$) | Ice Ablation Rate-Measure (m w.e.d$^{-1}$) |
|---|---|---|---|---|
| 8−6−2013 to 8−13−2013 | 0.049 | 0.045 | 0.064 | 0.073 |
| 7−27−2014 to 8−7−2014 | 0.055 | 0.053 | 0.075 | 0.077 |
| 7−29−2015 to 8−4−2015 | 0.057 | 0.053 | 0.075 | 0.078 |
| 7−25−2016 to 7−30−2016 | 0.053 | 0.054 | None | None |
| 8−12−2016 to 8−21−2016 | 0.048 | 0.050 | 0.070 | 0.067 |
| 7−31−2017 to 8−12−2017 | 0.056 | 0.060 | 0.084 | 0.078 |
| 8−5−2018 to 8−10−2018 | 0.061 | 0.051 | 0.072 | None |
| 8−4−2019 to 8−9−2019 | 0.051 | 0.051 | 0.072 | 0.073 |
| 7−29−2020 to 8−5−2020 | 0.055 | 0.053 | 0.075 | None |

**Table 3.** Impact of 24 heat events on North Fork Nooksack (NFN) and South Fork Nooksack discharge and temperature, glacier ablation, glacier runoff, and overall glacier contribution to flow of NFN.

| Start Date | End Date | NFK Discharge (%) | SFK Discharge (%) | NFK Temp (°C) | SFK Temp (°C) | NFK Glacier Ablation (md$^{-1}$) | Glacier Discharge (m$^3$s$^{-1}$) | NFK Discharge (m$^3$s$^{-1}$) | Glacier Runoff (%) |
|---|---|---|---|---|---|---|---|---|---|
| 7/20/09 | 8/5/09 | 50% | −34% | 1 | 4.7 | 0.058 | 8.46 | 28.05 | 30% |
| 7/25/10 | 8/1/10 | 7% | −25% | 0.3 | 1.0 | 0.048 | 7.00 | 41.7 | 17% |
| 8/14/10 | 8/19/10 | 19% | −14% | 0.7 | 1.8 | 0.055 | 8.02 | 29.2 | 27% |
| 9/5/11 | 9/14/11 | 30% | −8% | 0.4 | 1.2 | 0.055 | 8.02 | 24.2 | 33% |
| 7/7/12 | 7/14/12 | 40% | −29% | 0.2 | 1.5 | 0.057 | 8.31 | 88.6 | 9% |
| 8/11/12 | 8/19/12 | 18% | −16% | 0.3 | 1.5 | 0.063 | 9.18 | 35.3 | 26% |
| 8/6/13 | 8/13/13 | 15% | −14% | 0.6 | 0.7 | 0.045 | 6.56 | 25.5 | 26% |
| 7/7/14 | 7/18/14 | 13% | −29% | 2.1 | 4.7 | 0.054 | 7.87 | 45.9 | 17% |
| 7/27/14 | 8/7/14 | 5% | −44% | 0.8 | 2.7 | 0.053 | 7.72 | 25.3 | 31% |
| 6/25/15 | 7/21/15 | 53% | −30% | 0.7 | 3.9 | 0.051 | 7.26 | 18.8 | 39% |
| 7/29/15 | 8/4/15 | 16% | −16% | 0.6 | 1.4 | 0.053 | 7.54 | 13 | 58% |
| 7/25/16 | 7/30/16 | 10% | −11% | 0.8 | 1.6 | 0.054 | 7.69 | 24.2 | 32% |
| 8/12/16 | 8/21/16 | 18% | −20% | 0.7 | 1.7 | 0.05 | 7.12 | 17.5 | 41% |
| 7/31/17 | 8/12/17 | 13% | −22% | 0.5 | | 0.06 | 8.26 | 21.8 | 38% |
| 8/26/17 | 9/8/17 | 20% | −11% | 1 | 0 | 0.055 | 7.57 | 16.4 | 46% |
| 7/12/18 | 7/18/18 | 4% | −16% | 0.6 | 1.4 | 0.053 | 7.30 | 29.3 | 25% |
| 7/22/18 | 8/2/18 | 19% | −18% | 1 | 4 | 0.057 | 7.85 | 25.6 | 31% |
| 8/5/18 | 8/10/18 | 11% | −9% | 0.6 | 2.1 | 0.059 | 8.12 | 22.4 | 36% |
| 8/14/18 | 8/23/18 | 2% | −19% | 0.5 | 1 | 0.052 | 7.16 | 19.3 | 37% |
| 8/4/19 | 8/9/19 | 10% | −20% | 0.6 | 1.2 | 0.051 | 7.02 | 17.3 | 41% |
| 7/26/20 | 8/3/20 | 47% | −21% | 0.6 | 2.6 | 0.053 | 7.30 | 26.6 | 27% |
| 8/15/20 | 8/20/20 | 69% | −12% | 1.5 | 4.2 | 0.059 | 8.13 | 17.2 | 47% |
| 6/25/21 | 7/1/21 | 27% | −21% | 0.8 | 2 | 0.072 | 9.92 | 73.1 | 14% |
| 7/26/21 | 8/6/21 | 22% | −19% | 0.3 | 1.2 | 0.055 | 7.58 | 26.3 | 34% |
| Avg. | | 24% | −20% | 0.7 | 2.0 | 0.055 | 7.66 | 27.08 | 32% |

### 4.3. Nooksack River Discharge and Stream Temperature

The 24 heat events are noted in Table 3. The change in discharge is reported as the percentage change in discharge from the start of the heat event to the maximum or minimum discharge during the event at the USGS gages in both NFN and SFN. For the NFN, discharge increased by more than 10% during 20 of the 24 periods, with a mean increase of 23%. In the SFN, discharge decreased by more than 10% during 22 of the 24 periods (Figure 4).

The stream temperature change is the difference between the daily stream temperature at the beginning of the period to the maximum daily temperature during the heat period at the USGS gage in both the NFN and SFN. Stream temperature rose by more than 1 °C during 5 of the 24 events in the NFN and during 21 of 23 events on SFN, temperature data were missing for one event for SFN (Figure 5). The mean stream temperatures change was 0.7 °C in NFN and 2.0 °C in SFN, quantifying the ameliorating impact of glaciers on stream temperature in NFN.

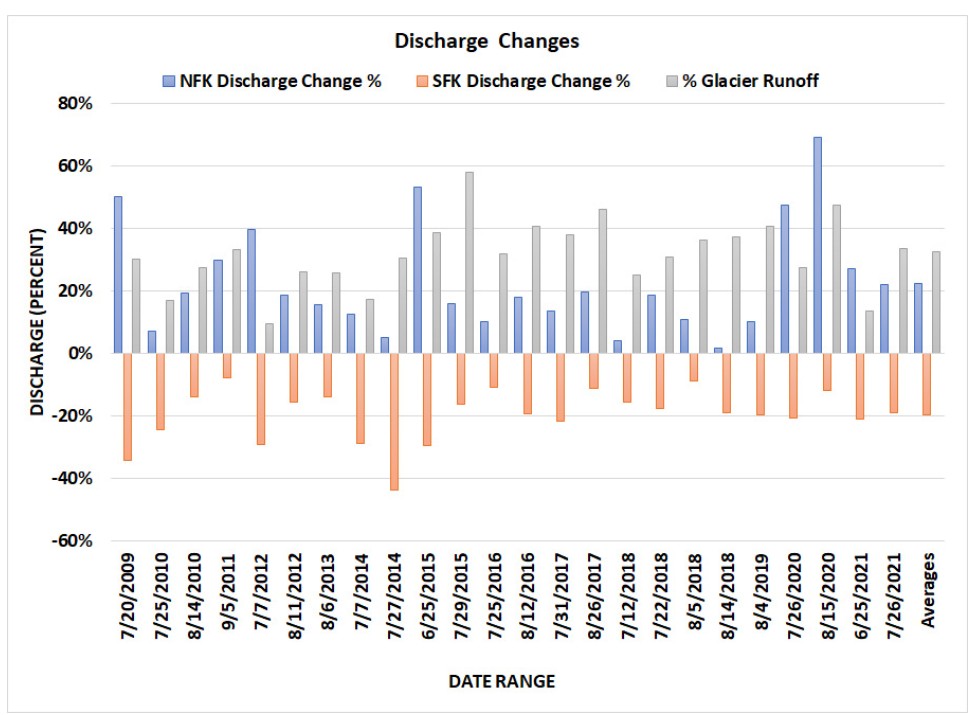

**Figure 4.** Change in discharge in the North Fork Nooksack (NFK) and South Fork Nooksack (SFK) during the 23 heat events. The percent of North Fork Nooksack discharge generated by glacier runoff is also indicated.

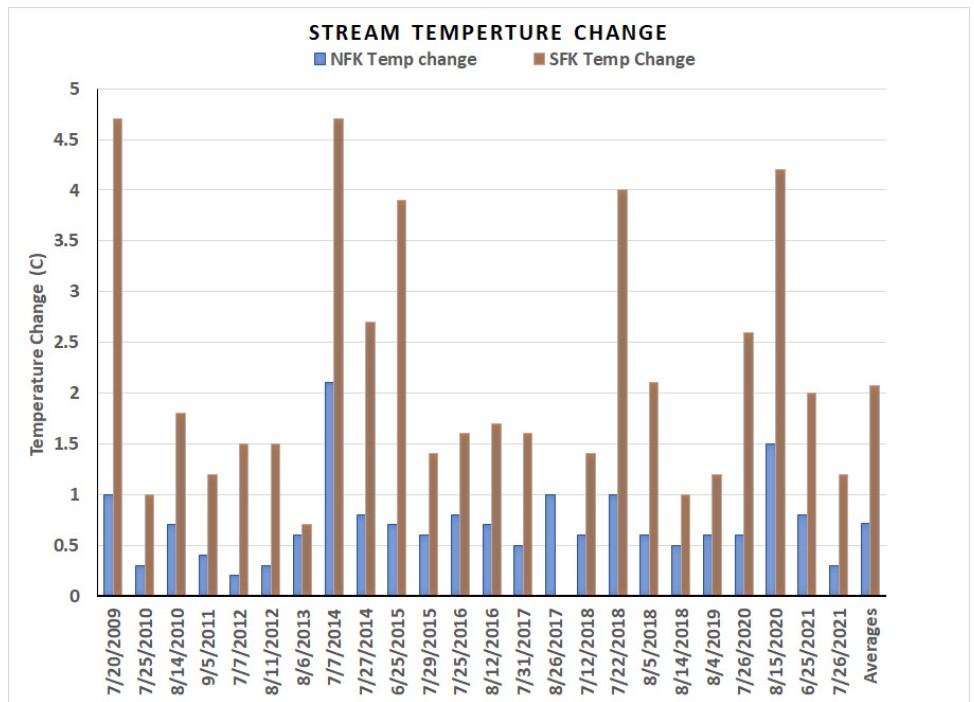

**Figure 5.** Change in daily stream temperature in the North Fork Nooksack (NFK) and South Fork Nooksack (SFK) during the 23 heat events from the beginning of the period to the maximum observed daily temperature.

The product of the observed or modelled daily glacier ablation and glacier area yields daily glacier discharge generated. Ablation measurements and modelling during these

events indicate a mean daily ablation ranging from 0.045 md$^{-1}$ w.e. to 0.063 md$^{-1}$ w.e., with a mean rate of 0.055 md$^{-1}$ w.e. The daily glacier discharge generated is then reported as a percent of the total observed NFN discharge. The daily melt is not all conveyed downstream to the gage the same day, hence the comparison to daily discharge is to glacier runoff generated. This analysis indicates that glacier runoff during these heat events generated discharge equivalent to more than 15% of total river flow during 22 of 23 events, with a mean of 34% (Figure 4; Table 3). Two of the events generated over 50% of the river discharge, both occurring during periods of particularly low flow.

*4.4. ERA5 Temperature Data*

The summer daily maximum temperature and daily lapse rate generated from ERA5 reanalysis for the 1979–2021 period yield a mean summer maximum temperature during 1979–2021 of 1.93 °C and a mean lapse rate of −5.73 °C km$^{-1}$. Maximum temperatures of greater than 10 °C approximates the 14 °C mean daily temperature threshold at the Middle Fork SNOTEL station for heat waves identification. From 1979–2021 there have been 378 days of the 6500 total days exceeding this threshold with an average lapse rate of −6.3 °C km$^{-1}$. During this period, 13 of the 20 highest maximum daily temperatures have been reported since 2015, with six of them (30%) occurring in 2021 (Table 4). This included a 141 h period from 25 June to 2 July 2021 where the temperature remained above 12 °C at 3000 m.

**Table 4.** The twenty highest maximum summer days with the highest maximum temperature at 3000 m from the ERA5 reconstruction for Mount Baker, Washington for May 1979 to July 2021. The lapse rate (°C km$^{-1}$) is also reported.

| Date | Lapse Rate (°C km$^{-1}$) | Maximum Temperature (°C) |
|---|---|---|
| 6/30/2021 | −5.99 | 18.88 |
| 6/29/2021 | −7.89 | 18.66 |
| 6/28/2021 | −7.47 | 17.55 |
| 7/1/2021 | −4.12 | 15.86 |
| 9/5/1988 | −6.71 | 15.53 |
| 7/13/2002 | −6.54 | 15.33 |
| 6/27/2021 | −6.53 | 15.33 |
| 9/4/1988 | −7.87 | 14.95 |
| 7/22/2006 | −7.10 | 14.86 |
| 9/3/1988 | −7.70 | 14.49 |
| 9/7/2017 | −6.14 | 14.39 |
| 6/28/2015 | −7.67 | 14.36 |
| 9/5/2017 | −7.53 | 14.33 |
| 6/26/2021 | −5.87 | 14.29 |
| 9/6/2017 | −7.13 | 14.12 |
| 8/10/2018 | −6.66 | 14.04 |
| 9/23/2009 | −6.54 | 13.81 |
| 6/27/2015 | −6.50 | 13.70 |
| 5/29/1983 | −8.29 | 13.64 |
| 7/31/2020 | −7.24 | 13.36 |

## 5. Discussion

In a glaciated watershed, glaciers are important to maintaining sufficient discharge and stream temperature that are critical for salmon populations. This is illustrated in the NFN where the 24 heat events have led to an increased discharge and a stream temperature rise of less than 1 °C. The increased discharge in NFN during heat waves while SFN discharge decreased demonstrates the impact of glaciers on the NFN reversing the discharge trend during heat events. This is the result of increased glacier ablation during the heat waves. The continued loss of glacier area will lead to a decline in this mitigating effect of glaciers on NFN stream conditions. How will this impact fish species?

Some cold-water trout and salmon species are already constrained by warm water temperatures and additional warming will result in net habitat loss [4,14,26]. In the Fraser River and Thompson River, British Columbia fish community thresholds were observed for mean weekly average temperatures of about 12 °C and again above 19 °C [26]. Below 12 °C, the community were characterized by bull trout and some cold-water species, between 12 °C and 19 °C by salmonids and sculpins, and above 19 °C by minnows and some cold-water salmonids [31]. The temperature threshold above which mortality increases markedly for Pacific salmon in the region is 15 °C [32,33]. These thresholds indicated that small temperature changes can be expected to drive substantial changes in fish communities. During the 24 heat events noted in the North Fork only two events exceeded 12 °C, while in the South Fork 14 of the events exceeded 19 °C, which is well above the threshold where mortality increases [32,33]. This suggests that both rivers are near a threshold that could alter the fish community composition.

In Pacific Northwest rivers, air temperature drives 82–94% of the long-term stream temperature trends [4,34]. Summer discharge and air temperature both account for approximately half of the inter-annual variation in stream temperatures [34,35]. In spring, no temperature increase was observed and the rate of warming was highest during the summer at 0.17–0.22 °C increase in temperature per decade [4].

Nooksack River salmon begin and end their life cycle in the Nooksack River. The Washington Department of Fish and Wildlife [36] SalmonScape project maps the distribution of salmon in the Nooksack River basin. Each population is mapped separately for spawning, rearing, and presence. Chinook, Coho, and Chum salmon in the North Fork can migrate up to the base of Nooksack Falls 40 km upstream of the NFN-Nooksack River Junction. The SFN has the most extensive network of salmon streams with the presence of salmon extending 52 km upstream of the junction of SFN with the Nooksack River (Figure 6).

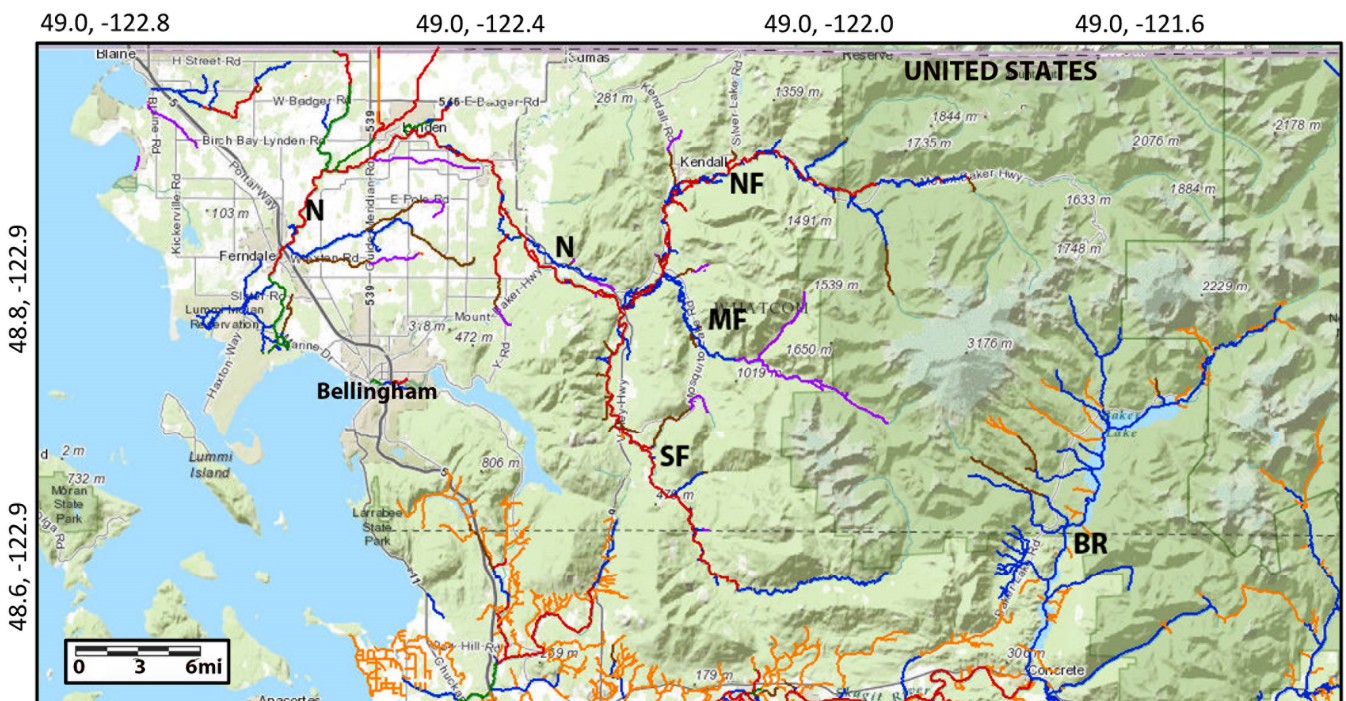

**Figure 6.** From the WDFW SalmonScape, this map indicates the extent of Chinook salmon in the Nooksack (N) and Baker River (BR) watersheds. Including the Nooksack sub-basins; MF = Middle Fork, NF = North Fork, SF = South Fork. Red = documented spawning, Blue = Documented presence, Green = Documented rearing, Yellow = Modelled presence, Purple = Blocked.

Chinook salmon surveys in the Nooksack River are conducted annually by the Washington Department of Fish and Wildlife. In the NFN Chinook spawn mainly in a 30 km

stretch from Mosquito Lake Road to Wells Creek at the base of Nooksack Falls [36]. In the NFN, the number of returning Chinook is divided into natural and hatchery spawned salmon; the WDFW [37] report that 88% of recent spawning Chinook salmon originate from the Kendall Creek Hatchery at the junction of Kendall Creek and the NFN. From 2000 to 2011 the number of Chinook released in the Middle Fork and NFN watershed averaged 1,036,000 sub-yearling fish [37]. Though overall populations and escapements increased as a result, natural-origin spawning Chinooks have not increased from 1995 to 2016 and remained threatened in the NFN River [37].

For salmon, both their riverine and marine environments are experiencing physical changes due to climate change, compounding human alteration of the aquatic habitat. This is a consistent stress throughout their life cycle [14,32,33]. In late summer of 2021, the SFN experienced a ~2500 Chinook die off from warm water lowering resistance to columnaris disease [38]. There was no die off in the NFN potentially indicative of the ameliorating impact of glaciers on stressful stream conditions.

## 6. Conclusions

The increasing frequency and intensity of Pacific Northwest heat waves underscores the need to quantify the impact on all alpine watersheds; in this case the Nooksack Basin a glaciated alpine watershed. Alpine glaciers in the NFN drive an increase in discharge during heat events averaging 24%, while limiting water temperature rise to a mean of 0.7 C. This contrasts with the unglaciated SFN where during the same heat events discharge declined 20% and temperatures increased 2.1 C. During the heat events increased ablation drove an increase in glacier runoff and the importance of glacier runoff to overall river discharge. Heat events are of importance, because the low discharge and high temperatures that characterize heat events are stressful for salmon populations.

Mass balance losses in the basin are driving glacier area decline [3], that has already led to a declining glacier runoff [3,8,19]. The result of continued glacier area loss will be a reduction in the enhanced discharge, leading to reduced flow during warm-dry low flow events. Ongoing loss of glacier area will also lead to a greater increase in overall stream temperature in NFN. The summer of 2021 brought the highest observed air temperatures to the region further highlighting the importance of this issue [39]. The ERA5 maximum temperatures identified that the three hottest summer days of the 1979–2021 period were 28, 29 June, and 30 June 2021.

This study is the first detailed quantification of glacier ablation, glacier runoff, and consequent alpine river discharge during heat waves in this region. The study highlights the importance of completing additional ablation measurements of bare ice surfaces and consistent repeat mapping of the distribution of snow-covered area on these glaciers using remote sensing products to effectively apply melt models.

**Author Contributions:** Data from field observations of glacier mass balance and ablation observation completed by M.S.P., J.P. and M.D. All weather station data, USGS discharge and steam temperature data were analyzed by M.S.P. ERA5 temperature reconstruction and lapse rate calculation was performed by T.M. and L.B.P.; writing—original draft preparation, M.S.P.; writing—review and editing by T.M. and M.D. All authors have read and agreed to the published version of the manuscript.

**Funding:** This research received no external funding.

**Data Availability Statement:** All USGS data used is available for North Fork Nooksack River is from: https://waterdata.usgs.gov/nwis/uv?site_no=12205000. South Fork Nooksack River data is from https://waterdata.usgs.gov/nwis/uv?site_no=12210000. Temperature data for the USDA Snotel site Middle Fork Nooksack is at: https://www.nwrfc.noaa.gov/snow/snowplot.cgi?MNOW1. Mass balance data is reported to the World Glacier Monitoring Service at: https://wgms.ch/data_databaseversions/ (all accessed on 28 January 2022).

**Acknowledgments:** This work has benefitted from the support of the Nooksack Indian Tribe. This project has been sustained by over 60 field assistants including several who spent at least five field seasons with the project Tom Hammond, Ben Pelto, and Bill Prater. The leadership in glacier monitor-

ing and data reporting of the World Glacier Monitoring Service has been integral to the ongoing work of the North Cascade Glacier Climate Project field program. The long-term discharge measurements by the USGS and weather data from the USDA SNOTEL system have been indispensable.

**Conflicts of Interest:** The authors declare no conflict of interest.

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
