# Peer review of "Contribution of Glacier Runoff during Heat Waves in the Nooksack River Basin USA"

_water, doi:10.3390/w14071145_

Round 1
Reviewer 1 Report
- Novelty of the paper is not clear.
- Please show the location of gauging stations in map.
- Table 3 and figure 4, figure 5, what is the threshold/baseline for discharge change, for eg. 40% change to what?
- How glacier runoff contribution is estimated?
- Please present the daily ablation (continuous time step) for the time period as shown in table 2.
- Content in chapter 4.1 is not the result. They shall be discussed in literature review or discussions.
Author Response
Attached are detailed comments for Reviewer 1

Reviewer 2 Report
Contribution of Glacier Runoff during heatwaves in the Nooksack River Basin USA
The study estimates the contribution and changes in glacier runoff under the heatwave conditions in the Nooksack River Basin USA. The study has used the Degree-Day Model for ablation modeling. Such studies are positively welcome to understand the impact of climate variability especially the climatic extremities that have increased significantly over the last few decades on regional hydrology. The manuscript is well-structured, however, to be beneficial for the readership of the “Water” journal I would suggest the following changes/clarification in the manuscript.
Main comments
The authors are requested to briefly describe the Degree-Day model, especially the calculation of the degree-day factors.
- The basis of the DDF used in the manuscript is not clear. For example, the statement at L197 indicates that the DDF was derived in the present study from in situ melt measurements, however, at L241 the authors have mentioned other references for the DDF of Ice. The authors need to discuss the DDF computation upfront in the manuscript.
- Also, the determination and differentiation of the glacier melt contribution in the river discharge are not clear. For example, how were the base flow and glacier runoff contributions distinguished?
- A few sentences on the comparison/accuracy of the reanalysis data must have been included in the manuscript.
- Furthermore, the authors are requested to provide uncertainty estimates of both the in situ and modeled mass balance measurements.
Further, Comments
L11-L12: Since the manuscript does not discuss the long-term mass balance changes the statement made here needs to be revised.
L24-L25: Why such a huge variation in the uncertainty of discharge data in NF and SF.
L28: “……further decreases in summer base-flow ” should it be a surface runoff?
L223: Please provide a figure or modify Figure 1 to depict the stake distribution.
Table 2: Please provide the uncertainty estimates for the ablation/mass balance measurements.
Table 3: Similarly provide the uncertainty estimates of the table entries including ablation and glacier runoff.
Include recent references on the topic.
Author Response
Attached are detailed responses to Reviewer 2. Thank you for comments that have improved the paper.

Reviewer 3 Report
The paper analyzes changes in the glacier ablation and its contribution to runoff during summer heat waves recorded in the Nooksack River Basin, USA. In my opinion, the research is interesting and deserves attention. However, the article requires some improvements prior to its final acceptance for publication in the journal, especially in terms of:
- supplementary information about the study area, including its hydrological and climatic characteristics, and their long-term variability,
- methodology, including the definition of “heat wave”, and criteria of establishing threshold values applied in the study,
- editorial improvements, including the quality of figures, the editing of tables, and the notation of units.
Detailed comments are as follows:
- Keywords: please remove “glacier runoff” and “Nooksack River” from keywords as they already appear in the manuscript title.
- Figure 1: the scale bars are in “miles” (Mount Baker glaciers) and “kilometers” (Nooksack River Watershed) – please unify the unit notation. What is the meaning of the yellow dots (gauges?) and the red line shown in the upper figure? Please add relevant explanations. Moreover, please add relevant labels to the USGS stations. There is meteorological station is not marked in the figure.
- Table 1: “Mean Elevation m” – please add “m a.s.l.” (meters above sea level) – the same with elevations mentioned in the text (for example, on p. 5, l. 167-168). Moreover, in Table 1 please add elevation for the Nooksack River. In the first column title add “Basin”, and in the last column change “2008-13” into “2008-2013”.
- Methods and data sources:
- how is the term “heat wave” defined in your study? Please add relevant explanation (definition),
- p. 6, l. 198: please explain how the threshold values for temperature and precipitation were determined in your study.
- Results:
- please check if the units are written correctly, for example, on p. 6, l .231-232: “0.55 m w.e..”, while on p. 9, l. 289: “0.045 md-1 w.e.”,
- Table 2: please unify the date notation, and also unify the “Ablation Rate” to three decimals,
- Figure 2: please improve its quality (blurred, and thus hardly readable), and also improve the notation of units on the horizontal [° C] and vertical [m e.d-1] axes,
- Figure 3: in my opinion the bottom photo is unnecessary and can be deleted,
- Table 3: is too narrow and needs to be “stretched”.
- Discussion:
- Figure 6: please add a frame with the geographical coordinates and a linear scale,
- please discuss the results of your study with reference to the results obtained by other researchers
- Conclusions:
- please avoid repeating the contents of the “Results” chapter.
Author Response
Attached are detailed responses to Reviewer 3. Thank you for comments that have improved the paper.

Reviewer 4 Report
The paper is interesting for the scientific audience and I will recommend it publishing after some major revisions
Introduction
I think a clear statement of the key point of the research is currently missing from the last paragraph. A generic description is presented.
Methodology
There is lack of methodology applied. Details of the dataset are provided however I would recommend to see a descriptive paragraph on the works carried out. In this current format there is a jump from materials/method to the results section. Need to eliminate this gap such as justify DDF . Also I understand the requirement of using temperature ERA5 data however there is no discussion of the dataset quality.
Results
See my above comment on the reliability of ERA5 temperatures associated with the findings of chapter 4.4.
Discussion
Seems to be a conclusion chapter without presence of limitation of the study, further research needs etc.
Author Response
Attached are detailed responses to Reviewer 4. Thank you for comments that have improved the paper.

Round 2
Reviewer 1 Report
Comments are addressed and thus recommended for acceptance
Author Response
Dear Reviewer:
Thank you very much for your comments.
Reviewer 2 Report
Thanks for addressing the comments.
Author Response

(The authors gave the same response as above.)

Reviewer 3 Report
Explanations and corrections made by the Authors are satisfactory. I recommend to accept the paper for publication in present form.
Author Response

(The authors gave the same response as above.)

Reviewer 4 Report
I'm happy with author's responses and I would recommend it for publication
Author Response

(The authors gave the same response as above.)
